# Association between Head-to-Chest Circumference Ratio and Intrauterine Growth-Retardation Related Outcomes during Preweaning and Postweaning

**DOI:** 10.3390/ani12121562

**Published:** 2022-06-17

**Authors:** Diego Manriquez, Guilhem Poudevigne, Etienne Roche, Agnes Waret-Szkuta

**Affiliations:** Centre de Coopération Internationale en Recherche Agronomique Pour le Développement (CIRAD), Animals, Health, Territories, Risks, Ecosystems (ASTRE), École Nationale Vétérinaire de Toulouse (ENVT), Université de Toulouse, 23 Chemin des Capelles, 31300 Toulouse, France; diego.manriquez-alvarez@envt.fr (D.M.); guilhem.poudevigne@envt.fr (G.P.); e.roche@socsa.fr (E.R.)

**Keywords:** growth, immaturity, PCV-2, sow prolificacy

## Abstract

**Simple Summary:**

Large litter size might cause significant variation in piglet intrauterine development and growth. Piglets affected by intrauterine growth-retardation (IUGR) have abnormal head shapes and body conformations, as well as lower survival, growth, and meat quality. Therefore, methods for discriminating lower growth and performance piglets that will support the management of IUGR at the individual and farm levels are needed. We hypothesize that piglets with lower birth weight, colostrum intake, average daily gain, and immune response against PCV-2 have differential head-to-chest circumference ratios (HCRs) at birth. We observed that greater HCRs were associated with lower birth weight, colostrum intake, and weight gain. Additionally, piglets with greater HCRs had higher PCV-2 antibodies. The HCR was associated with outcomes linked to IUGR. Thus, HCR might be used as an objective, low-invasive, and inexpensive tool to assess newborn piglets and assist in neonatal and nursery management.

**Abstract:**

The objective of this study is to evaluate the association between the head-to-chest circumference ratio (HCR) and birth weight (BW), colostrum intake, and average daily weight gain (ADG) at preweaning and postweaning periods. Additionally, associations between HCR and PCV-2 serum antibody titers and the PCV-2 seroconversion ratio (SCR) were assessed. Head and chest circumferences were measured at birth, and HCR was calculated from 110 piglets born from 8 pregnant sows randomly selected from maternity pens. Linear mixed models were used to test whether changes in HCR were associated with fluctuations of BW, colostrum intake, and ADG. In addition, HCR least-square means were compared between piglets classified as lower or greater BW, colostrum intake, and ADG. Finally, receiving operating characteristic curve analyses were performed to estimate HCR thresholds for discriminating between lower and greater performance piglets during preweaning and postweaning periods. Increments in HCR were associated with lower BW, colostrum intake, and ADG. An HCR threshold of 0.82 maximized sensibility and specificity for the classification of lower and greater performance piglets regarding BW, colostrum intake, and ADG during the periods of 0 to 7 and 0 to 69 days of life. When piglets were categorized into HCR ≤ 0.82 and HCR > 0.82 groups, piglets with HCR ≤ 0.82 had lower (log10) PCV-2 serum antibody titers at 26 days of life compared with piglets with HCR > 0.82 (3.30 ± 0.05 vs. 3.47 ± 0.05 g/dL). On the other hand, piglets that showed low SCR between 26 and 69 days of life had greater HCRs compared with piglets with high SCRs (0.83 ± 0.008 vs. 0.8 ± 0.008). The use of HCRs allowed us to identify piglets with lower performance and impaired immune response against PCV-2. The HCR indicator could be used as a selection criterion for preventive management for piglets showing delayed performance potentially associated with IUGR.

## 1. Introduction

The number of piglets weaned per litter is an important indicator of the profitability of swine operations, and it is the most important indicator of sow productivity [1,2,3]. Nonetheless, current evidence suggests that a larger litter size also increases the proportion of piglets with lower birth weight and colostrum intake, as well as impaired suckling behavior, gut adaptation, and survival [1,4,5,6], which are associated with piglets’ exposure to intrauterine growth-retardation (IUGR). This disorder has become a major problem for the swine industry, affecting approximately 5% to 10% of pregnancies [1,7,8,9]. Additionally, the occurrence of IUGR is associated with higher incidences of preweaning morbidity and mortality, impaired animal welfare, and long-term effects on growth and development [10,11]. 

Traditionally, detection of IUGR piglets relies on the classification of piglets using the birth weight (BW) fifth and tenth percentiles or 1.5 standard deviations below the litter BW mean. In addition, a **visual assessment of head morphology** has been included to define IUGR, **in which** dolphin-like heads, characterized by a steep forehead, bulging eyes, and wrinkles perpendicular to the mouth, are a feature [3].

Piglet IUGR has been attributed to varied causes, including intrauterine anatomic and physiologic triggers. A crowded gravid uterus affects placental functions regarding size, vascularity, and surface area, producing reduced blood flow and nutrient supply, which causes heterogeneous birth weights and body development in large litters [12,13]. By 90 days of gestation, piglets with IUGR exhibit a reduction in secondary muscle fibers and in the mass of fetal organs relative to brain size [13]. This results in less myofiber hyperplasia, fatter carcasses, and lower meat quality compared with heavier piglets within the litter [1]. In addition, piglets affected by IUGR show lower immune organ weights in the thymus, spleen, and mesenteric lymph nodes, as well as a reduced number of goblet cells and lymphocytes and decreased gene expression and levels of cytokines [6,14]. Thus, the assessment of piglets with impaired intrauterine growth at an early age may have a pivotal role in the success of vaccinations against important clinical conditions, such as **diseases associated with** porcine circovirus type 2 (PCV-2), and in the success of strategies to improve the overall immune status, from birth to finishing.

The extent of piglet intrauterine development is associated with important economic traits. Birth weight has been correlated with survival, carcass and meat quality, and growth efficiency during fattening [15,16,17]. Nonetheless, assessing IUGR based only on body weight may overlook critical considerations of embryonic development established early in gestation [18]. In response to this issue, morphometrics and facial features have been associated with IUGR and lower performance. For instance, a positive genetic correlation between chest circumference and backfat thickness, as well as body weight, has been observed in Duroc pigs [19]. Moreover, the organ weights of piglets suffering IUGR are significantly lower compared with normal piglets [1]. Additionally, IUGR piglets show signs of gut adaptation delay, altered microbiota, and intestinal disease [13,20,21,22].

Morphometrics is a valuable tool to determine the variability of intrauterine growth and the potential effects of IUGR on piglet performance. Objective morphometrics can assist in the selection against IUGR phenotypes and in the implementation of preventive programs [23]. Thus, it will be critical to validate methods for assessing variability in IUGR-related outcomes within a litter in an efficient, quantitative, inexpensive, and easy way to manage and prevent this detrimental condition. To that end, we hypothesize that a head-to-chest circumference ratio (HCR) can estimate the extent of brain-to-lung ratio fetal development and is associated with preweaning and postweaning performance, in which greater HCRs are detrimental to productive outcomes. The objective of this study is to evaluate the association of HCRs with birth weight (BW), colostrum intake, average daily weight gain (ADG), and PCV-2 serum titers and seroconversion. Additionally, we assess the effect of covariates measured during the nursery period on the study outcomes. 

## 2. Materials and Methods

### 2.1. Study Design, Farm, and Animals

A total of 118 piglets (females *n* = 66; males *n* = 52) born from 8 sows randomly selected from a farrow-to-finish commercial farm located in southwestern France were enrolled in a prospective single cohort study from birth until 69 days of life. Primiparous sows were not included. The study farm housed 375 LW × Landrace sows (PIC genetics, Hendersonville, TN, USA) grouped in 10 pens of 32 animals. The terminal boars were Pietran. Pregnant sows are vaccinated with Rhiniseng^®^ (Hipra, Amer, Spain), Suiseng^®^ (Hipra, Amer, Spain), and Respiporc Flu3^®^ (Ceva, Libourne, France). Lactating sows are vaccinated with Parvoruvax^®^ (Ceva, Libourne, France).

In the study farm, the average number of weaned piglets per sow was 12.5, and the postweaning facility had the capacity for 1700 piglets. At birth, piglets were identified using button ear tags (Chevillot^®^, Albi, France). Maternity facilities included five rooms with 10 to 20 places. The reproductive management considered farrowing every other week. The study farm had 1800 postweaning places and 3900 fattening places. All rations were fabricated on the farm.

The study piglets were born between 12 and 14 February 2019 and were housed in two different farrowing rooms with comparable conditions. At 20 or 22 days of life, the study piglets were transferred to postweaning pens according to the farm’s management protocols. The dimensions of the postweaning pens were 7 m^2^, with a maximum capacity for 34 piglets. Piglets enrolled in the study received 2 doses of 0.5 mL I.M of Circovac^®^ (Ceva, Libourne, France) and Stellamune Mycoplasma^®^ at 26 and 47 days of life. Feeding and health management were performed under the farm’s protocols.

### 2.2. Study Outcomes

This study comprised birth weight (BW, g), colostrum intake (g), average daily gain (ADG, g/day) during preweaning and postweaning periods; PCV-2 antibody titers (log10 g/dL) at 26, 47, and 69 days of life; and PCV-2 seroconversion ratios (SCRs) as the study outcome variables. These variables were prospectively collected, as shown in Figure 1.

To test the association between HCRs and the study outcomes, two analytic approaches were performed. First, BW, colostrum intake, and ADG were analyzed as continuous variables to assess the effect of changes in HCRs on the study outcomes. Second, BW, colostrum intake, ADG, PCV-2 titers, and SCRs were categorized as lower (levels below first distribution quartile) and greater (above first quartile) to assess the means of HCRs and study the potential classification thresholds of piglets with a lower performance by using receiving operating characteristic curve (ROC) analyses.

Study piglets were weighed at birth and at 24 h of life using a bucket and a hanging digital scale (HBD 5K5N, Kern & Sohn, Ebingen, Germany). The BW was estimated according to the procedure suggested by Launay (2018) [24], deducting the estimated weight of the umbilical cord. Colostrum intake was estimated from the weight gain at 24 h of life using the colostrum intake equation developed by Devillers et al. (2007) and Declerck et al. (2016) [25,26]. The ADG was estimated during the preweaning and postweaning periods. Individual values were calculated for periods between 0 to 7, 7 to 21, 21 to 47, and 47 and 69 days of life. Additionally, we calculated the overall ADG during the study period, comprising 0 to 69 days of life.

The PCV-2 antibody titers at days 26, 47, and 69 of life **and SCRs** were categorized according to interquartile distribution into low, medium, and high levels, as proposed by Figueras-Gourgues et al. (2019) [27]. The SCR was calculated by dividing the log10 PCV-2 titer of a later time point by an earlier time point. This ratio estimated the percentual change of PCV-2 antibodies, in which values above 1 indicate increments, whereas values below 1 indicate decrements. The SCRs were calculated for time points between 47 and 26, 69 and 26, and 69 and 47 days of life.

### 2.3. Explanatory Variables

The main explanatory variable of this study is the head-to-chest circumference ratio (HCR). Other covariates of interest are the piglet sex, birth vitality score, birth time, birth order, cross-fostering, and teat latching categories. 

The HCR was considered a continuous variable in the linear regression analyses of the study outcomes. However, for the analysis of PCV-2 antibody titers, piglets were categorized into two groups: HCR ≤ 0.82 and HCR > 0.82.

The birth vitality of piglets was scored according to the procedures proposed by Baxter et al. (2007) [28]. According to the vitality assessed during the first 15 s of life, three categories of piglet vitality were considered: piglets that breathed but did not move, piglets that breathed and moved, and piglets that breathed, moved, and attempted to stand up. We recorded the birth time of the study piglets related to the first-born piglet in each litter. Consequently, three categories of piglets were considered for the analyses: piglets born within 1 h, piglets born between 1 and 2 h, and piglets delivered after 2 h. In addition, the birth order was recorded. Three categories of piglets were considered for the analyses: piglets born within the first and fifth place, piglets born between the sixth and tenth place, and piglets from the eleventh place. 

Movements of lactating piglets to fostering sows were analyzed. A farm manager determined piglet cross-fostering, and the research team did not interfere with these decisions. Four categories of cross-fostering movements were considered for analyses, piglets that were not cross-fostered, piglets cross-fostered within 6 h of life, piglets cross-fostered between 6 and 24 h of life, and piglets cross-fostered after 24 h of life. Finally, observations of the piglets’ teat preference were performed between 7 and 10 days of life, and a teat latching category was created. This category was classified according to piglets nursing from thoracic (teats 1 to 4), abdominal (teats 5 to 10), and inguinal (teats 11 to 16) teats.

### 2.4. Morphometrics

Individual measurements of HCRs were collected within one hour of birth. Circumferences of the head at the eye level and of the chest at the elbow level were measured with measuring tape (cm), as depicted in Figure 2. 

### 2.5. Blood Sampling and Measurement of PCV-2 Antibodies

Blood samples were collected at 1, 26, 47, and 69 days of life by puncture of the jugular vein using sterile 20-gauge and 2.5 cm length needles. Blood samples were allowed to clot in blood collection tubes, and serum was harvested and stored until laboratory analysis.

Serum samples for determination of the levels of PCV-2 antibodies were submitted to LABOCEA 22 (Ploufragan, France) for quantitative ELISA (SERELISA^®^ PCV2, Zoetis, Parsippany, NJ, USA). 

### 2.6. Statistical Analyses

Data editing and analyses were performed in RStudio (Version 2021.09.2). Linear mixed models were used to test the association between the study outcomes and covariates. Sow ID was considered a random effect. For model building, univariate analysis was performed between each study outcome variable and the study covariates. Covariates with univariate associations with *p*-values < 0.15 were included in the initial models. Additionally, we tested collinearity in covariates selected for the initial models using the chi-squared test. Backward elimination was used to select the final models for each outcome variable. Covariates with *p*-value < 0.1 were retained in the final models for confounder control. Linear regression estimates were used to determine associations between continuous covariates and outcomes, whereas ANOVA was used to determine differences between categorical variables. Thresholds of HCR were calculated using logistic regression and receiving operating characteristic curve (ROC) analyses. Study outcomes were categorized into the lower or greater category according to quartile distribution, and HCRs were used as the continuous variable in the logistic regression models. Classification tables were generated, and probability levels that maximized sensitivity and specificity were selected for threshold calculation. Statistical significance was set at a *p*-value <0.05.

## 3. Results

### 3.1. Descriptive Statistics

Mean litter size was 14.75 piglets/sow (SD = 1.9; min = 12; max = 16). The mean sow parity number was 3.5. A total of 110 piglets (females *n* = 62, males *n* = 48) completed the prospective study. Seven piglets died during the lactation period, and one piglet died during the nursery period. Data from dead piglets were excluded from the analyses. Table 1 shows the overall means and standard deviations of the study outcomes. Concerning explanatory variables, the overall HCR mean was 0.82 (SD = 0.04; min = 0.70; max = 0.92). Proportions of piglets allocated into the covariate categories are presented in Figure 3. Table 2 presents linear regression estimates of the associations between HCRs and the outcome variables of this study. All models were adjusted by significant and controlling covariates, which are presented in each outcome section below. 

### 3.2. Birth Weight and Colostrum Intake

We assessed the association between HCRs and other covariates with birth weight (BW) and colostrum intake. When we analyzed BW as a continuous outcome variable, only HCR (*p* < 0.0001) and sex were significant predictors. This implies that increments of HCR were associated with lower BW, as indicated by the negative sign of the HCR estimate (Table 2). Thus, the interpretation from this estimate indicates that for every 0.01-unit of HCR increase, BW decreases by 33.1 g. Regarding sex, male piglets had greater BW compared with female piglets (LSM ± SEM, 1511.5 ± 37.1 vs. 1403.4 ± 33.7 g, *p* = 0.015).

After the categorization of BW, 27 (24.5%) piglets were categorized as lower BW (≤1282.3 g) and 83 (75.5%) as greater BW (>1282.3 g). Piglets in the lower BW category had greater HCRs compared piglets with greater BW (0.85 ± 0.008 vs. 0.81 ± 0.006; *p* < 0.0001). The ROC analysis indicated that an HCR threshold of 0.83 can discriminate between lower BW and greater BW piglets with 83.1% sensitivity and 74.1% specificity (area under the curve (AUC) = 0.85).

To study colostrum intake as a continuous variable, HCR, BW category, and birth vitality score were selected to build the colostrum intake model. However, the birth vitality score did not remain in the final model due to its *p*-value (*p* = 0.37). In the final model, we determined that HCR was associated with colostrum intake (Table 2) after controlling by BW category. Thus, greater HCRs are associated with lower colostrum consumption. Accordingly, increments of HCR in 0.01-units were linked with a decrease in colostrum intake of 6.9 g. Regarding BW category, piglets in the lower BW category had lower colostrum intake compared with piglets with greater BW (308.2 ± 22.2 vs. 374.8 ± 16.3 g; *p* = 0.002).

We observed that piglets in the lower colostrum intake (≤300.7 g) category (*n* = 27, 24.5%) had greater HCRs compared with piglets in the greater colostrum intake (>300.7 g) category (*n* = 83, 75.5% (0.84 ± 0.009 vs. 0.81 ± 0.006 g; *p* = 0.0002)). With this association, we determined that an HCR threshold of 0.83 can classify lower vs. greater colostrum intake piglets with 73.5% sensitivity and 63% specificity (AUC = 0.77).

### 3.3. Average Daily Weight Gain

We studied the effect of HCR and other covariates on the average daily weight gain (ADG, g/day) during preweaning and postweaning periods. Additionally, we included the BW and colostrum intake categories as controlling variables because they are commonly reported as important factors for ADG. Table 2 shows the linear association between HCR and ADG as a continuous outcome.

In all ADG models (0 to 7, 7 to 21, 21 to 47, and 47 to 69 days of life), HCR had a negative estimate slope, which implies that increments of HCR were associated with decrements of ADG (Table 2). For ADG measured between 0 to 7 days of life, we determined that HCR, colostrum intake category, birth vitality score, cross-fostering, and teat latching categories were significant predictors. Thus, increments of HCR in 0.01-points were associated with ADG decrements of 2.9 g/day (Table 2). Within the covariates, piglets classified with lower colostrum intake had lower ADG scores compared with those with greater colostrum consumption (150.4 ± 9.5 vs. 185.5 ± 6.8 g/d; *p* = 0.0008). We did not find differences between birth vitality score categories in the LSM comparisons; however, they were retained in the model as a controlling variable. 

Cross-fostering had a significant effect on ADG between 0 to 7 days of life. Piglets that were not cross-fostered had greater ADG compared with piglets cross-fostered after 24 h of life (196.3 ± 6.1 vs. 115.1 ± 16.5 g/day; *p* < 0.0001). Additionally, piglets cross-fostered within 6 h of life (178.2 ± 10.5 g/day; *p* = 0.006) and those cross-fostered between 6 and 24 h of life (182.3 ± 11.9 g/day; *p* = 0.006) had greater ADG compared with piglets cross-fostered after 24 h of life. 

The teat latching category was associated with ADG at 0 to 7 days of life. Piglets nursing from the thoracic teats had greater ADG compared with those milking from the inguinal teats (185.0 ± 8.17 vs. 150.68 ± 10.8 g/day; *p* = 0.008). We did not observe differences in ADG between piglets latched on thoracic and abdominal (168.2 ± 7.3 g/day) or abdominal and inguinal teats.

When we classified piglets as lower (≤157.8 g/day; *n* = 28 (25.4%)) and greater (>157.8 g/day; *n* = 82 (74.5%)) ADG during the period of 0 to 7 days of life, we determined that lower ADG piglets had greater HCRs compared with greater ADG piglets (0.84 ± 0.009 vs. 0.81 ± 0.007; *p* = 0.0004). The ROC analyses determined that a 0.82 HCR threshold can be used to discriminate between lower and greater ADG during this period with 63.4% sensitivity and 64.3% specificity (AUC = 0.71).

Increments in HCR resulted in lower ADG during the period of 7 to 21 days of life (Table 2), in which a 0.01-unit increase of HCR decreased ADG by 2.4 g/day. Besides HCR, the significant predictors of ADG at 7 to 21 days of life were the BW, birth vitality score, and cross-fostering categories. There was no difference between lower and greater BW piglets; however, this category was retained as a controlling variable. Interestingly, piglets with lower birth vitality scores (breathed but not moved within 15 s after birth) had greater ADG during the period of 7 to 21 days of life compared with piglets that breathed and moved (260.9 ± 13.8 vs. 233.1 ± 11.5 g/day; *p* = 0.04). On the other hand, we did not observe differences between piglets that breathed, moved, and tried to stand up (245.9 ± 15.7 g/day) and the other birth vitality scores. Concerning cross-fostering and its effect on ADG during the period of 7 to 21 days of life, we determined that piglets that were not cross-fostered had greater ADG compared with piglets cross-fostered within the first 6 h of life (272.8 ± 11 vs. 229.6 ± 15.8 g/day; *p* = 0.03); we did not observe other differences among other cross-fostering categories (6–24 h, 235.8 ± 18.8; >24 h, 248.3 ± 21.7). There was no association between HCR and the lower (≤226.2 g/day; *n* = 28 (25.4%)) and greater (>226.2 g/day; *n* = 82 (74.5%)) ADG categories, measured between 7 and 21 days of life, and HCR (0.83 ± 0.01 vs. 0.81 ± 0.007; *p* = 0.16); therefore, we did not perform ROC analyses.

Only HCR and the colostrum intake category were retained in the final model of ADG between 21 and 47 days of life. Nonetheless, HCR was not associated with ADG (*p* = 0.09, Table 2). On the other hand, colostrum intake was a significant predictor of ADG during this period. In consequence, piglets with lower colostrum intake had lower ADG during the period of 21 and 47 days of life compared with piglets with greater colostrum intake (244.5 ± 22.8 vs. 290.6 ± 18.8 g/day; *p* = 0.01). 

The analysis of ADG between 47 and 69 days of life showed that HCR was not a significant predictor for this period (Table 2). Birth weight, birth order, and cross-fostering (*p* = 0.1) categories remained in the final model. Piglets in the lower BW category had lower ADG compared with those classified as greater BW (490.8 ± 32.0 vs. 555.5 ± 25.5 g/day; *p* = 0.04). Regarding the birth order category, piglets born within the 5th place had greater ADG compared to those born within the 6th and 10th place (548.7 ± 27.2 vs. 483.4 ± 27.9; *p* = 0.02), whereas no differences were observed between piglets born after the 10th place (527.3 ± 33.4 g/day) and other birth order groups. As expected, we did not find differences in HCR between the lower and greater ADG categories during the period of 47 to 69 days of life (0.83 ± 0.01 vs. 0.81 ± 0.007; *p* = 0.18). 

Finally, we assessed the effects of HCR and other covariates on ADG during the whole study period (0 to 69 days in life). The HCR, BW, and cross-fostering categories were retained in the final model. The cross-fostering category remained a controlling variable. Thus, 0.01-point increments of HCR were associated with 2.3 g/day decreases of ADG between 0 and 69 days of life (Table 2). Piglets in the lower BW category had lower ADG compared with piglets in the greater BW category (327.2 ± 18.9 vs. 372.2 ± 16.3 g/day; *p* = 0.004). We allocated 28 (25.4%, ≤337.7 g/day) piglets in the lower ADG category and 82 (75.6%, >337.7 g/day) in the greater ADG category between 0 to 69 days of life. After this categorization, we observed that lower ADG piglets had greater HCRs compared with greater ADG piglets (0.84 ± 0.009 vs. 0.81 ± 0.007; *p* = 0.01). Consequently, we determined that a 0.83 HCR threshold yields a 71% sensitivity and 53.6% specificity (AUC = 0.67) for discriminating ADG performance during the period of 0 to 69 days of life.

### 3.4. PCV-2 Antibody Titers and Seroconversion

Overall means (SD) of PCV-2 antibody titers and the PCV-2 seroconversion ratio (SCR) are presented in Table 1. For analyzing the PCV-2 and PCV-2 SCR, we considered HCR as a categorical variable (HCR ≤ 0.82 and HCR > 0.82). After this categorization, 58 (52.7%) piglets were included in the HCR ≤ 0.82 group, whereas 52 (47.3%) were classified in the HCR > 0.82 group. Univariate analyses for PCV-2 included the study’s explanatory variables and piglet colostrum intake, sow colostrum IgG concentration (g/dL), and piglet serum IgG concentration (g/dL).

The PCV-2 antibody titers measured at 26 days of life were associated with the HCR category, whereas birth time, colostrum intake, and sow colostrum IgG were included as controlling variables. Piglets in the HCR ≤ 0.82 group had lower PCV-2 antibody titers compared with piglets classified as HCR > 0.82 (3.30 ± 0.05 vs. 3.47 ± 0.05 g/dL; *p* = 0.02). We did not find associations between PCV-2 antibodies measured at 47 or 69 days of life. Only birth vitality score and teat latching categories were predictors of PCV-2 titers at 69 days of life. Thus, piglets that breathed, moved, and tried to stand up had greater PCV-2 antibody titers compared with piglets that only breathed and moved (3.89 ± 0.06 vs. 3.66 ± 0.04 log10 g/dL; *p* = 0.006) and with piglets that only breathed during the first 15 s after birth (3.89 ± 0.06 vs. 3.61 ± 0.05 log10 g/dL; *p* = 0.002). Piglets latched to inguinal teats had greater antibody titers compared with piglets feeding from abdominal teats (3.81 ± 0.07 vs. 3.59 ± 0.04 log10 g/dL; *p* = 0.01), whereas piglets milking from the thoracic teats tended to have greater antibody titers compared with piglets milking from the abdominal teats (3.74 ± 0.05 vs. 3.59 ± 0.04 log10 g/dL; *p* = 0.06). 

We were interested in the relationship between HCRs and SCRs during periods corresponding to 26 to 47, 26 to 69, and 47 to 69 days of life. Categories were created as low, medium, or high based on the quartile distributions of SCRs. For SCRs between 26 and 47 days of life, a total of 27 (25.5%) piglets were classified as low SCR (≤0.92), 52 (49%) piglets as medium SCR (>0.92 to ≤1.02), and 27 (22.5%) piglets as high SCR (>1.02). For SCRs between 26 and 69 days of life, a total of 27 (25.5%) piglets were classified as low SCR (≤0.96), 52 (49%) piglets as medium SCR (>0.96 to ≤1.22), and 27 (22.5%) piglets as high SCR (>1.22). Finally, for SCRs between 47 and 69 days of life, a total of 27 (24.8%) piglets were classified as low SCR (≤1.01), 54 (49.5%) piglets as medium SCR (>1.01 to ≤1.19 units), and 28 (25.7%) piglets as high SCR (>1.22).

We observed an association between HCRs and SCRs at 26 to 69 days of life. Thus, piglets classified in the low category had significantly greater HCRs compared with piglets in the high category (0.83 ± 0.008 vs. 0.80 ± 0.008; *p* = 0.02), whereas piglets in the low category tended to have greater HCRs compared with piglets in the medium category (0.83 ± 0.008 vs. 0.82 ± 0.006; *p* = 0.1). Lastly, we did not observe an association between HCRs and SCRs between 26 and 47 or 47 and 69 days of life.

## 4. Discussion

In this study, we aimed to integrate morphometrics measured in vivo and to assess the effect of changes in HCRs on important performance indicators during preweaning and postweaning periods. Current evidence suggests that the morphology of body parts and organ weights are associated with performance and survival, which are potentially related to IUGR. For instance, separate measurements of head and chest circumferences have been proposed as relevant indicators of economic traits in pigs because they can estimate growth and physiological status [19]. Additionally, greater abdominal circumference has been associated with greater ADG during the postweaning period [29]. Regarding organ weights, it has been reported that piglets showing signs of IUGR have smaller hearts, livers, and kidneys [3,7].

To control IUGR, several approaches have been developed, including nutritional interventions during gestation [8,30] and selection management at the maternal and piglet levels [23]. The results of these approaches have been contradictory and, to some extent, controversial. On one hand, studies have concluded that energy denser diets for pregnant sows increase piglets’ muscular tone at birth and weaning weight; however, postweaning growth and vitality were not improved [8,30]. On the other hand, breeding selection against IUGR traits, which have low heritability, and for sows that deliver heavier piglets in smaller litters have been proposed [23,31], although this may trigger a potentially unsustainable decrease in litter size. At the piglet level, there is extensive research on management to overcome the negative effects of IUGR and improve piglet vitality and survival. Examples include feeding supplementation, maternity crates designed to avoid crushes and cold stress, and supplementation with IGF-1, EGF, and polyphenols as gut protectors [9,30]. However, the success of these strategies depends heavily on the ability to detect piglets with signs of IUGR accurately and efficiently. Therefore, there is a critical need for developing and validating in vivo measurements associated with performance outcomes that are also linked to IUGR. These metrics could support strategies for improving piglet health and performance and for evaluating IUGR farm status. 

Intrauterine growth-retardation has been measured using BW and facial features commonly found in IUGR piglets, including dolphin-like heads, characterized by a steep forehead, bulging eyes, and wrinkles [3]. However, these approaches would also include small piglets with symmetric morphology that are not necessarily exposed to IUGR [18]. Some studies have demonstrated that piglets affected by IUGR have distinct body conformations and organ development. One study determined that piglets born with the aforementioned facial characteristics had lower BW, ADG, and survival and a higher brain-to-organs size ratio. However, the determination of head morphology is performed visually [3], which may introduce human error into piglet classification. In our study, we used an objective measurement of the head circumference (Figure 2) that can be repeated consistently. Nonetheless, none of these methods can explain all sources of variability in outcomes related to performance and immaturity by themselves. In the case of the calculated HCR thresholds, lower sensitivity and specificity were obtained in ADG measured at a later age. Despite differences in methodology, studies have agreed that piglets exposed to IUGR have abnormal head morphology and that head shape assessment could be an efficient management tool to evaluate IUGR [29,32].

Similar to our approach, some studies have used ratios of body part measurements to estimate the magnitude of IUGR and the subsequent effect on body weight and vitality. Brain-to-liver, -heart, and -BW ratios, as well as the head-circumference-to-BW ratio, have been explored [3,7], in which head circumferences were measured at 9 weeks of age. To the best of our knowledge, there are no other studies that have explored the use of HCRs at birth to estimate piglet performance during the preweaning and postweaning periods. With the evidence presented here, we suggest that HCRs can estimate the fetal development of the brain and thoracic organs from live newborn piglets regarding organ weight. This is important because the brain develops early during gestation, while the lung matures at a later gestation age [33]. This is referred to as the brain-sparing effect, and it is distinctive in piglets suffering from IUGR, showing dolphin-like head shapes [34]. Thus, the fetus may exhibit asymmetrical growth, which is characterized by a normal growth of the brain and the restricted growth of other organs such as lung, liver, and heart [35]. Therefore, greater HCRs can be expected in IUGR piglets, as we have determined in this study. 

The use of HCRs as a continuous variable is an advantage compared with IUGR measures based on the categorization of visual observations of body shape. Although studies have been able to determine IUGR through visual judgment and the categorization of BW [3,6,7,20], our approach might be valuable in determining subclinical IUGR. Remarkably, the linear mixed model estimates presented in our study allow us to estimate BW, colostrum intake, and ADG, considering other important factors that occur during the first week of life that are also associated with IUGR [28,36]. In this sense, we observed similar results in other studies in which greater brain-to-organ weights and brain-to-BW ratios were a sign of IUGR and were linked to reduced preweaning and postweaning performance [3,13]. 

There is evidence that BW and colostrum intake are associated with IUGR and that they are determinants of piglet performance [3,26]. In agreement with this idea, the ADG estimation models were adjusted by controlling for BW and colostrum intake categories that might confound the observed associations between HCR and ADG.

Birth time and birth order have been considered relevant factors for ADG. Contrary to our results, one study found that increments in birth order were associated with increments of ADG, although in that study, birth order was not associated with other variables such as vitality and mortality [26]. We determined that birth time, colostrum intake, and sow colostrum IgG should be controlled when estimating serum antibody titers of PCV-2. Other studies have reported that birth order explains some of the variability of the IgG serum concentration of piglets due to access to colostrum supply [37]; however, its role as a predictor of piglet performance remains unclear [26,38]. On the other hand, birth order has been considered a relevant predictor of piglet mortality [28,37]. Nonetheless, our study setting only allowed us to determine its effect on ADG. 

It is reasonable that covariates related to piglet vitality, cross-fostering, and teat latching are interrelated. For this reason, we tested their interactions in our models; however, no interactions were found. In this study, HCR and birth vitality score were significant factors explaining the variability of ADG during the preweaning and postweaning periods. Our results agreed with other studies, concluding that lower vitality piglets have lower BW and ADG [28]. Cross-fostering is a practice performed in up to 98% of pig farms [39]. In our study, the decision to allocate piglets to foster sows was a relevant factor associated with ADG; hence, it was a relevant indicator for performance outcomes, which are related to IUGR. This might open opportunities to use HCRs to support cross-fostering decisions with an objective metric. The only period when cross-fostering was not associated with ADG was between 21 to 47 days of life. Nonetheless, as observed in other studies, piglets that remained with their dams had better overall performance [39,40,41].

Piglets establish hierarchies for nursing from specific teats, and these preferences become stable during the first 7 days of life and remain stable during lactation [34,42]. Our results agreed with other studies that found that piglets latched to thoracic teats are heavier and gain more weight [37,43]. Studies have confirmed better ADG during the first month of life in piglets lactating from the thoracic teats [42]. Contrastingly, we observed that piglets nursing from thoracic teats only had greater ADG during the period of 0 to 7 days of life. This might be explained by the stabilization of weight gain after teat selection across all study piglets. 

Regarding the PCV-2 serum antibodies, we observed that at 69 days of life, piglets milking from inguinal teats had higher antibodies compared with those latched on abdominal teats. This is interesting because differences were detected when the peak of antibody levels was expected, around 80 days of life [44]. This might be because piglets milking from inguinal teats had the least interference of maternal antibodies [27]; however, more research is needed to test this hypothesis.

Study limitations include the use of a single farm; however, this farm was managed under the conventional practices of pig farms in France. Additionally, we acknowledge that only between 32% to 46% of the variability is explained by our models. Therefore, there are unmeasured sources of variations in our study that will need to be identified and measured.

## 5. Conclusions

This study indicates that greater HCRs are associated with lower BW and colostrum intake. We observed an association between HCR and ADG during the preweaning and postweaning periods after adjusting for BW. Piglets with greater HCRs show a lower ADG. In addition, a 0.82–0.83 HCR threshold was able to classify lower and greater performance piglets with moderate to high sensitivity and specificity. Piglets with greater HCRs had lower PCV-2 SCRs. HCRs could be used as a selection criterion for piglets showing signs of IUGR at the farm level and to implement preventive management with a population approach. Birth vitality score, cross-fostering, and teat latching were relevant covariates associated with outcomes linked to piglet immaturity.

## Figures and Tables

**Figure 1 animals-12-01562-f001:**
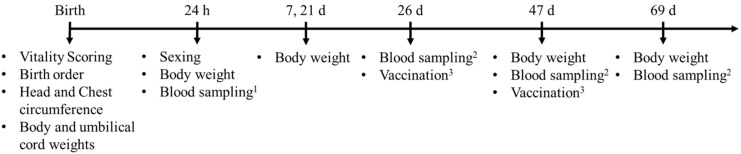
Timeline of the study procedures performed during preweaning and postweaning periods. Blood samples for determination of serum IgG ^1^ and PCV-2 ^2^ antibodies titer. h: hours of life; d: days of life. ^3^ Vaccination against PCV-2. Clinical inspection was performed daily throughout the study period.

**Figure 2 animals-12-01562-f002:**
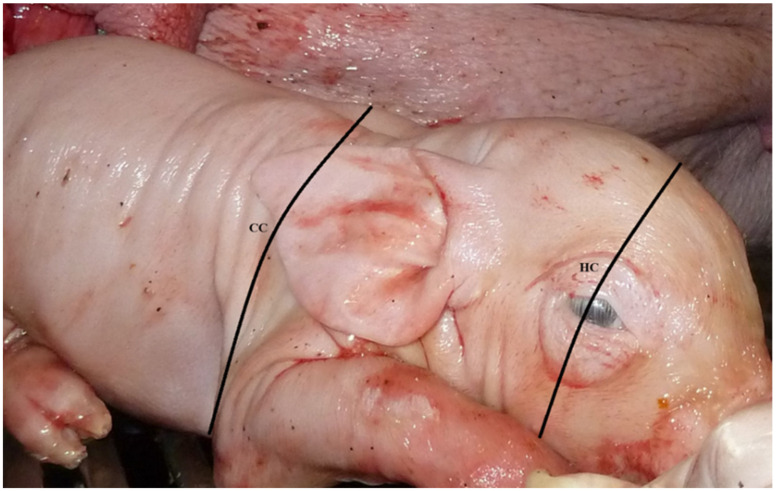
Measuring points of head circumference (HC) and chest circumference (CC) from newborn piglets. Photo copyright by Guy-Pierre Martineau.

**Figure 3 animals-12-01562-f003:**
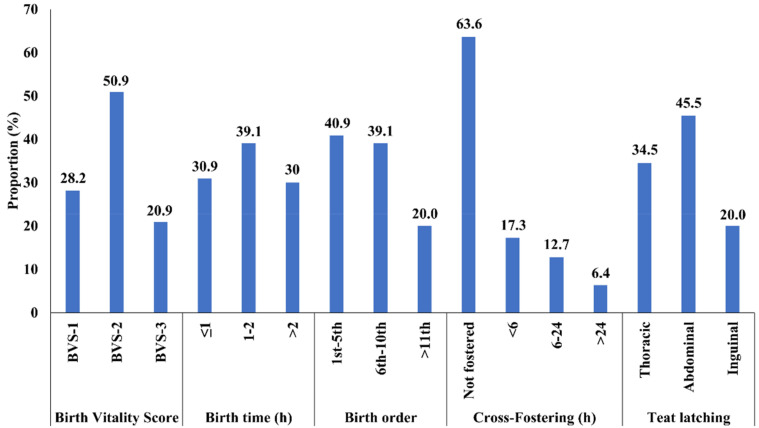
Proportion of study piglets (*n* = 110) classified in the covariates measured during the preweaning and postweaning periods. BVS1: breathed and not moved during the first 15 s of life; BVS-2: breathed and moved within 15 s of life; BVS-3: breathed, moved, and tried to stand up during the first seconds of life. Birth time (h) relative to the first piglet born. Cross-fostering (h) after birth.

**Table 1 animals-12-01562-t001:** Mean and standard deviation (SD) of the study outcome variables measured in piglets followed from birth until 69 days of life.

Outcome Variables	Mean	SD
Birth weight (g)	1451.6	274.9
Colostrum intake (g)	357.6	101.9
Average daily gain (g/day)		
0–69 d	374.0	71.6
0–7 d	196.7	54.6
7–21 d	261.5	57.5
21–47 d	282.1	87.3
47–69 d	549.6	126.0
PCV-2 antibody titers (log10)		
26 d	3.4	0.4
47 d	3.3	0.3
69 d	3.7	0.3
PCV-2 SCR ^1^		
47–26 d	1.0	0.1
69–26 d	1.1	0.2
69–47 d	1.1	0.2

^1^ PCV-2, seroconversion ratio (SCR) between sampling points. SD, standard deviation.

**Table 2 animals-12-01562-t002:** Adjusted HCR linear regression estimates associated with study outcomes.

Study Outcomes	Estimate	SE	*p*-Value	Adj. R-Squared
Birth weight (g)				
Intercept	4114.3	429.3	<0.0001	0.34
HCR	−3314.6	521.7	<0.0001	
Colostrum intake (g)				0.39
Intercept	870.1	189.6	<0.0001	
HCR	−687.2	220.9	0.0002	
Average daily gain (g/day)				0.46
0 to 7 days				
Intercept	444.9	87.9	<0.0001	
HCR	−292.8	102.2	0.005	
7 to 21 days				0.34
Intercept	445.2	119.1	0.005	
HCR	−204.4	138.2	0.005	
21 to 47 days				
Intercept	498.9	156.5	0.002	
HCR	−311.1	183.8	0.09	
47 to 69 days				0.32
Intercept	566.1	267.5	0.03	
HCR	−26.3	315.4	0.9	
0 to 69 days				0.46
Intercept	539.3	133.4	0.0001	
HCR	−232.9	155.1	0.0001	

## Data Availability

Data sets generated from this study are available upon request from the corresponding author.

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
