# Peer review of "Association between Head-to-Chest Circumference Ratio and Intrauterine Growth-Retardation Related Outcomes during Preweaning and Postweaning"

_animals, 2022, doi:10.3390/ani12121562_

Round 1
Reviewer 1 Report
Dear Authors,
looking for new methods, parameters which can help us evaluate animals, identify certain disorders and counteract adverse productive performance, are worth considering and researching. Because of it your research is interesting but presentation obtained results requires some changes.
In my opinion you should rewrite Introduction and Discussion parts and correct/improve a little description of Methods and Results.
The first part of discussion should be moved to the introduction. Decide what is your aim of the study. Do you want only to rely on evaluation the association of HCR with birth weight, colostrum intake, average daily weight gain (ADG), and PCV-2 serum titers and seroconversion - it is rather quite obvious. I understand that you want to combine it with IUGR piglets recognition and I didn't find satisfactory explanation. compare what you wrote in the Simple Summary and in the Introduction.
Piglets defined as IUGR pigs included the criteria: (a) a steep dolphin-like forehead, and one or more of the following characteristics: (b) bulging eyes, (c) wrinkles perpendicular to the mouth or (d) hair with no direction of growth. If none of the criteria applied, the piglet was defined as normal. We don't rely only on body weight - line 57, you have to give here some more explanation.
For me if you based only on HCR you can write only about normal and small and/or cachectic piglets. I expect that in group of smaller piglets according to your criteria (HCR>0.82) we could find some smaller and IUGR piglets - how many? Decide what would you want to show it is important to write correctly Discussion (in your version the Discussion does not discuss the results enough, it often just cites the results again or the literature, but does not discuss both).
Present the study design in details - were piglets divided after the first measurement into 2 groups?, did you note if/how many you have IUGR piglets?, whether these piglets included in the group of small piglets. What are exactly these small piglets?, did you based only on the HCR?, didn't you take into account their vitality? It should be presented more clearly.
Later you can change a little Table 1 and present there results for normal and small (perhaps additionally IUGR) piglets and of course results for whole population could be included.
After you respond to the comments presented above, I can review your work.
Additionally check once again whole text, I found there some small mistakes like for instance: double use BO-2 (l.155) or Figure 1.
Author Response
Dear Reviewer,
We thank you for your comments and insights and for your time in reviewing our manuscript. We are convinced your input has improved the clarity and the value of our research. We have responded to all your concerns to the best of our knowledge. Changes are included in the manuscript and specific responses are included below
- AU: indicate our responses below each comment
Sincerely,
The authors.
Dear Authors,
looking for new methods, parameters which can help us evaluate animals, identify certain disorders and counteract adverse productive performance, are worth considering and researching. Because of it your research is interesting but presentation obtained results requires some changes.
In my opinion you should rewrite Introduction and Discussion parts and correct/improve a little description of Methods and Results.
- AU: We have reviewed and modified the introduction and discussion sections according to your overall insights and the inclusion of methodology and results. Thank you.
The first part of discussion should be moved to the introduction. Decide what is your aim of the study. Do you want only to rely on evaluation the association of HCR with birth weight, colostrum intake, average daily weight gain (ADG), and PCV-2 serum titers and seroconversion - it is rather quite obvious. I understand that you want to combine it with IUGR piglets recognition and I didn't find satisfactory explanation. compare what you wrote in the Simple Summary and in the Introduction.
- AU: Thanks for your comments. We have made the study objective more concise. This was to evaluate the association between HCR and the outcome variables. We acknowledged that with the available data, direct associations between HCR cannot be inferred.
Piglets defined as IUGR pigs included the criteria: (a) a steep dolphin-like forehead, and one or more of the following characteristics: (b) bulging eyes, (c) wrinkles perpendicular to the mouth or (d) hair with no direction of growth. If none of the criteria applied, the piglet was defined as normal. We don't rely only on body weight - line 57, you have to give here some more explanation.
- AU: We agree. We have expanded this point in the manuscript considering that the determination of IUGR is complemented by head morphology and body morphology as well as methods that measure fetal growth.
For me if you based only on HCR you can write only about normal and small and/or cachectic piglets. I expect that in group of smaller piglets according to your criteria (HCR>0.82) we could find some smaller and IUGR piglets - how many? Decide what would you want to show it is important to write correctly Discussion (in your version the Discussion does not discuss the results enough, it often just cites the results again or the literature, but does not discuss both).
- AU: Thank you for bringing this point up. We agree and we have limited our results to lower and greater performance piglets. Additionally, we performed ROC analyses to determine HCR thresholds associated with lower or greater performance. Distributions of piglets classified in these categories are included in the manuscript.
Present the study design in details - were piglets divided after the first measurement into 2 groups?, did you note if/how many you have IUGR piglets?, whether these piglets included in the group of small piglets. What are exactly these small piglets?, did you based only on the HCR?, didn't you take into account their vitality? It should be presented more clearly.
- AU: We have added more details about the study design. We have specified that this study is a prospective single cohort study. Researchers did not perform any grouping of the study piglets and no treatments or interventions were included based on piglet separation. Criteria for small piglets are presented in the materials and methods and in the result sections. Birth vitality score was included as a covariate.
Later you can change a little Table 1 and present there results for normal and small (perhaps additionally IUGR) piglets and of course results for whole population could be included.
- AU; We have changed the title of Table 1. This table was created to describe the overall dispersion of the outcome variables. Means of outcome variables are presented by lower and greater performance in the results section.
After you respond to the comments presented above, I can review your work.
Additionally check once again whole text, I found there some small mistakes like for instance: double use BO-2 (l.155) or Figure 1.
- AU: Thank you. Additionally, we have decided to significatively reduce the number of abbreviations.
Reviewer 2 Report
Results apply to a single farm. No information is provided on sow and pig feeds or on sow parity or vaccination that might assist comparison to other farms. No mention is made of piglet mortality or how this was handled in the analysis. Much more detail is needed in the methods to describe the farm environment and practices.
ADG from one time period to the next is highly correlated and this should be taken into account.
Conclusions would be much more useful if HCR and birthweight were compared as predictors of all outcomes. This would show whether HCR is a better predictor than birthweight.
A critical threshold for HCR should be determined by fitting inflection points, not by using the mean (unless 50% of piglets suffer from IUGR).
References should be reviewed to ensure accuracy.

Author Response
Dear Reviewer,
We thank you for your comments and insights and for your time in reviewing our manuscript. We are convinced your input has improved the clarity and the value of our research. We have responded to all your concerns to the best of our knowledge. Changes are included in the manuscript and specific responses are included below
- AU: indicate our responses below each comment
Sincerely,
The authors.
Results apply to a single farm. No information is provided on sow and pig feeds or on sow parity or vaccination that might assist comparison to other farms. No mention is made of piglet mortality or how this was handled in the analysis. Much more detail is needed in the methods to describe the farm environment and practices.
- AU: We agree with all your comments. We have added more details in the materials and methods and in the results sections. Information about the farm is included in a way it does not compromise farm confidentiality.
ADG from one time period to the next is highly correlated and this should be taken into account.
- AU: We agree. We considered adding previous ADG in the models, however, we observed a collinearity problem with HCR. For this reason, we added birth weight to the models to control for piglet size.
Conclusions would be much more useful if HCR and birthweight were compared as predictors of all outcomes. This would show whether HCR is a better predictor than birthweight.
- AU: Thank you for your comment. We have used both HCR and birth weight as predictors. Results from this approach are presented in the manuscript.
A critical threshold for HCR should be determined by fitting inflection points, not by using the mean (unless 50% of piglets suffer from IUGR).
- AU: We agree with your concern. To support the use of an HCR threshold we have included ROC analysis. This analysis supported an HCR threshold of 0.82 to 0.83.
References should be reviewed to ensure accuracy.
- AU: We have reviewed references accordingly.
Reviewer 3 Report
The manuscript entitled “Association between head to chest circumference ratio and intrauterine growth-retardation outcomes during pre and post-3 weaning” has been reviewed. The results showed that head and chest circumference ratio (HCR) was associated with growth, survival and immune responses of piglets. The manuscript is interesting. However, the results were not presented well. It would be better to convert some tables to graphical presentation for easy understanding of the readers.
Page 1 line 40 specie or species
Please avoid so many abbreviations in the manuscript. Each table or figure must be self-explanatory.
In discussion, first paragraph is not needed actually. A long background of the study was again stated at the beginning of discussion section. It would be better to shorten first and second paragraphs to make a single one. Discussion should be rewritten. Its better to highlight each of the results in each paragraph.
Author Response
Dear Reviewer,
We thank you for your comments and insights and for your time in reviewing our manuscript. We are convinced your input has improved the clarity and the value of our research. We have responded to all your concerns to the best of our knowledge. Changes are included in the manuscript and specific responses are included below
- AU: indicate our responses below each comment
Sincerely,
The authors.
The manuscript entitled “Association between head to chest circumference ratio and intrauterine growth-retardation outcomes during pre and post-3 weaning” has been reviewed. The results showed that head and chest circumference ratio (HCR) was associated with growth, survival and immune responses of piglets. The manuscript is interesting. However, the results were not presented well. It would be better to convert some tables to graphical presentation for easy understanding of the readers.
Page 1 line 40 specie or species
- AU: Thank you. Corrected in the manuscript.
Please avoid so many abbreviations in the manuscript. Each table or figure must be self-explanatory.
- AU: We agree. Thanks for your suggestion. We have substantially reduced the number of abbreviations. We have changed the title and table legends to be more self-explanatory. Also, we have converted table 2 into a table.
In discussion, first paragraph is not needed actually. A long background of the study was again stated at the beginning of discussion section. It would be better to shorten first and second paragraphs to make a single one. Discussion should be rewritten. It’s better to highlight each of the results in each paragraph.
- AU: Thank you for your suggestion. We have restructured the introduction and discussion according to your feedback.
Round 2
Reviewer 1 Report
Dear Authors,
I found a big differences in quality and form of results presentation. Now (I hope) I understand what authors wanted to describe and present. I have only a few comments for the authors to consider and perhaps worth to respond in the text.
- first I'm not a native speaker but still I found some language mistakes (plural/singular, present/past tense) - correct it;
- l.29: An?
- l.35-37 - this sentence is a slight abuse, no other antibodies were tested. I don't say that it is wrong ;)
- l.71-72 - what strategies?, however I found answer in discuss, perhaps it is worth to mentioned something here;
- Tab.1 : "colostrum intake", looking into title we don't know what value is presented it is worth to clarify it, I'm sure we have here suckling period up to about the 21-22nd day of life not 69, and it is an average for 1 piglet?
- l.266-270, what is it GP2?, suddenly I lost the concept. Above we have a descripton (legend) of the Fig.3 and sudden "jump" to GP2
- l. 346-347
Generally it was a pleasure to read it, now I like the description of results and discuss. For the future it is worth to evaluate more animals due to the high individual variability.
Good luck